# Vibration Induced Transport of Enclosed Droplets

**DOI:** 10.3390/mi10010069

**Published:** 2019-01-19

**Authors:** Hal R. Holmes, Karl F. Böhringer

**Affiliations:** 1Department of Bioengineering, University of Washington, Seattle, WA 98105, USA; hrholmes@uw.edu; 2Department of Electrical & Computer Engineering and Institute for Nano-engineered Systems (NanoES), University of Washington, Seattle, WA 98105, USA

**Keywords:** droplet, vibrations, transport, microfluidics

## Abstract

The droplet response to vibrations has been well characterized on open substrates, but microfluidic applications for droplets on open systems are limited by rapid evaporation rates and prone to environmental contamination. However, the response of enclosed droplets to vibration is less understood. Here, we investigate the effects of a dual-plate enclosure on droplet transport for the anisotropic ratchet conveyor system. This system uses an asymmetric pattern of hydrophilic rungs to transport droplets with an applied vibration. Through this work, we discovered that the addition of a substrate on top of the droplet, held in place with a 3D printed fixture, extends the functional frequency range for droplet transport and normalizes the device performance for droplets of different volumes. Furthermore, we found that the edge movements are anti-phasic between top and bottom substrates, providing a velocity profile that is correlated to vibration frequency, unlike the resonance-dependent profiles observed on open systems. These results expand the capabilities of this system, providing avenues for new applications and innovation, but also new insights for droplet mechanics in response to applied vibration.

## 1. Introduction

The response of liquid droplets to a vibration is typically studied with unconfined droplets on a homogenous substrate. In this configuration, researchers have provided thorough characterizations of resonance [1,2], contact angle hysteresis [3,4,5], and the movement of the contact line [6,7] in response to vibrations. Similar studies have also demonstrated that droplet transport can be induced by asymmetric vibrations [8,9], and sinusoidal vibrations can be converted to horizontal transport on gradient surfaces [10,11,12], creating the basis for microfluidic systems using these principles [13,14]. More recent work has leveraged this ability to rectify vertical vibrations into horizontal transport by creating patterned asymmetric surfaces that transport droplets over indefinite distances (e.g., transport does not stop at the end of a gradient) [14,15,16,17,18,19]. These systems present much potential to meet a variety of applications in microfluidics, but, for some applications (particularly those that require heating or long incubation times), are limited by their high evaporation rates compared to enclosed systems [20,21]. Therefore, this work demonstrates the implementation of an enclosed configuration for droplet transport based on the anisotropic ratchet conveyor system.

Anisotropic ratchet conveyors (ARCs) transport droplets through a periodic pattern of curved hydrophilic rungs, defined by a hydrophobic background. The curvature of these rungs divides the footprint of these droplets into leading and trailing edges, that experience a difference in pinning forces, as only the leading edge of the droplet conforms to the curved rungs (Figure 1) [14].

A sinusoidal vibration, applied vertically, causes the edges of the droplet to oscillate between wetting (edges of the droplet are advancing away from the centroid) and de-wetting (edges are contracting toward the centroid) phases. This combination of edge oscillation and a difference in pinning forces between leading and trailing edges results in a net force that drives droplet transport [14,18,19]. Previous work demonstrated a first principles model that showed droplet transport was enabled by two key anisotropies on ARCs: (1) pinning forces are greater on the leading edge of the droplet, and (2) the droplet is less susceptible to the ARC pattern during wetting (i.e., the difference in contact angle between the leading and trailing edges is larger during de-wetting than wetting) [22]. These characteristics provide for droplet transport with a ratcheting effect, wherein the droplet takes a small step backward and a larger step forward through each vibration cycle. From this foundation, additional functional capabilities of the ARC system have also been realized through the development of devices that can selectively pause, switch, and merge transported droplets [23], demonstrating the potential of ARC systems to automate sample handling protocols and processes. However, evaporation was still a major concern for this system. Although it was known that transport of an enclosed droplet is possible [24], the mechanics of vibrated droplets in an enclosed ARC system had yet to be investigated and it remained to be demonstrated that an enclosed ARC system could perform practical microfluidic tasks. In this work, a dual-plate ARC system was created to study how the addition of a top-plate enclosing and contacting the droplet affected the resulting transport of droplets. This system used 3D printed holders to align ARC substrates, allowing for the entire system to be vibrated. The results of this work advance the possibilities of the ARC system, providing new opportunities for innovation, and presenting new insights into droplet mechanics.

## 2. Materials and Methods

ARC fabrication begins by patterning a negative resist (NR9-1000PY) on an oxidized <100> p-type silicon wafer. Following development, the wafers are vapor coated with a hydrophobic silane (fluorooctyltrichlorosilane, FOTS, Sigma-Aldrich, St. Louis, MO, USA) to create the hydrophobic background (Figure 2). Wafers are then diced (Disco DAD321) into 25 × 75 mm plates. The photoresist is stripped from the ARC plates with acetone, revealing a transparent pattern of hydrophilic silicon dioxide (SiO_2_) rungs defined by the hydrophobic FOTS background [25]. Plate holders are fabricated through fused deposition molding with a 3D printer (Aleph Objects Lulzbot TAZ-6, Aleph Objects, Loveland, CO, USA) to provide a separation between the plates of 2.0 and 2.5 mm. Separation distances were confirmed through image analysis (Appendix A). This work characterized enclosed droplet transport with two configurations: (1) ARC-ARC, in which both top and bottom plates have the ARC pattern (i.e., they are mirrored), and (2) ARC-FOTS, in which the bottom plate has the ARC pattern and the top plate has a uniform hydrophobic FOTS coating.

ARC devices are driven by an electromagnetic motor (Brüel & Kjær, Nærum, Denmark) controlled by a function generator. The performance of the ARC devices is characterized by the minimum acceleration amplitude of the substrate required to initiate droplet transport (defined as the ARC threshold), which is measured with a laser Doppler vibrometer (Polytec OFV, Polytec Inc., Mooresville, NC, USA). This measurement was performed by increasing the vibration by small increments until transport occurs and recording the reading on the laser doppler vibrometer. The ARC threshold is reported in acceleration rather than displacement because, with a sinusoidal motion, the acceleration of the platform scales with the product of displacement and vibration frequency squared, and acceleration is directly proportional to the energy input of the system, as previously established [18,19,22,25]. The movement of droplets on the ARC devices is captured with a high-speed camera (Photron UX50, Photron, Tokyo, Japan), and these frames are analyzed in MATLAB (Matrix Laboratory, MathWorks, Natick, MA, USA) with custom scripts to provide quantitative measurements of transport velocity and edge movement. Error bars represent ± standard deviation in all quantitative data shown in this work.

## 3. Results

The configurations used in this work consisted of the ARC pattern on bottom and hydrophobic coating on top (ARC-FOTS) and the ARC pattern on both top and bottom (ARC-ARC) substrates. Separations of 2.0 and 2.5 mm were used for both configurations with 8, 13, and 18 µL droplets. Measurements of 8 µL droplets were only recorded with the 2.0 mm plate separation because 8 µL droplets did not contact the top plate when separated by 2.5 mm.

### 3.1. Effects of Top Plate Enclosure on Droplet Transport

Qualitatively, droplets transported in this dual-plate configuration appear pillarlike, with leading and trailing edges on both plates and a liquid bulge (the center of mass of the droplet) oscillating back and forth between the two plates (Figure 3A and Appendix A). ARC threshold measurements demonstrate that droplets transported in the dual-plate system require a larger vibration amplitude for transport at low frequencies than the open configuration, but at the upper end of the functional frequency range of the open plate system, a cross-over occurs, and the dual-plate system becomes more efficient (Figure 3B). On open ARC systems, the ARC threshold profile is characterized by a frequency where ARC threshold is a minimum (related to the physical properties of the droplet [22] and the pattern of the ARC device [25]) and a sharp increase in ARC threshold at the high frequency end of the functional frequency range. However, the ARC threshold profiles for the dual-plate configuration do not exhibit such pronounced increases in ARC threshold at the high frequency end. The functional frequency range for the dual-plate system is also much broader than that of the open system. As the separation distance is reduced, a similar trend occurs—the ARC threshold is higher at low frequencies but crosses over near the end of the functional frequency range of the dual-plate system with a larger separation. While droplets appeared more stable with the ARC-ARC configuration (de-pinning from the top plate was observed during some vibration cycles only with the ARC-FOTS system), no quantitative difference in performance was observed. These observations are elaborated in the discussion section.

### 3.2. Length Scale Normalization

Transport on the dual-plate system also appears to be independent of droplet volume. Plotting these measurements on the same axes for each separation distance shows aligned ARC threshold profiles for droplets of each volume interrogated (Figure 4). These data suggest that the dual-plate configuration normalizes the vertical movement of the center of mass of the droplets. In other words, for unconfined droplets in an open system, the vertical movement of the center of mass depends on the droplet volume [22] and vibration frequency [25], but on this enclosed system, the vertical movement of the center of mass is confined to a fixed range between the two plates.

### 3.3. Transport Velocity of Enclosed Droplets

The velocity of droplets transported under these configurations was also determined (Figure 5). Velocity was recorded at the ARC threshold, which therefore describes the minimum velocity that a droplet can be transported at for each frequency. Previous work examining droplet velocities in an open configuration showed that transport velocity profiles exhibited a large velocity at a specific frequency dependent on droplet volume. This velocity dropped as frequency was increased even though the acceleration (and therefore velocity) of the driving substrate was higher [25]. The peak in this velocity profile provided the appearance of a resonance frequency, but this frequency was higher than the predicted natural frequency of these droplets on a surface [6,22], and was influenced by the pattern of the ARC device [25]. While these peaks appear to be related to resonance of the droplet, additional theoretical modelling is required to more accurately describe these effects.

On the enclosed system, local maxima and minima were observed, but these peaks occurred at higher frequencies than on the open system and, generally, the transport velocity increased with increasing frequency. The velocity was also slightly higher on the ARC-FOTS configuration, which is likely due to the reduced pinning forces acting on the top of the droplet. This result is likely due to the tethering of the droplet to both top and bottom plates. In the dual-plate configuration, the droplet edges are wetting on the top plate and de-wetting on the bottom plate as the droplet center of mass moves upward and vice-versa as the center of mass moves downward. Thus, the movement of droplet edges is anti-phasic (Figure 6), so the leading edge on one plate cannot advance too far ahead of the other in any particular vibration cycle. Thus, unlike an open ARC system [25], the droplet consistently advances by the same number of rungs on each plate throughout each vibration cycle. Therefore, increasing frequency results in the droplets taking regular steps of the same distance more quickly (Appendix A).

## 4. Discussion and Conclusion

In previous work, droplet transport profiles on an open system were non-dimensionalized with respect to the radius of the droplet, as it was assumed that the displacement of the center of mass during vibration scaled with the radius of the droplet (i.e., the vertical oscillation of the center of mass gets larger as the droplets get larger) [22]. The alignment of ARC threshold profiles based on plate separation observed in this work provides further evidence for this scaling argument, as the dual-plate system effectively normalizes the displacement of the center of mass for each droplet. This effect also contributes to the broadening of the functional frequency range observed when the top plate is added and separation distance is reduced. The confined motion of the droplet center of mass emulates the motion of smaller droplets. Future work will look into creating a comprehensive model of these effects, as well as the effects of the droplet interaction with the top-plate. However, there are limits on this normalization effect, and measuring a broader range of droplet volumes would likely show a bandpass (or volume-pass) effect where droplets too small to contact the top-plate or too large to sufficiently respond to the ARC track will not be transported, but all volumes in between will exhibit a similar ARC threshold profile. This volume range is dependent on the plate separation distance and the feature size of the ARC pattern, and therefore could be selected for specific applications. For example, 8 µL droplets behave as open droplets while 13 and 18 µL droplets behave as confined droplets with a plate separation of 2.5 mm, meaning that 13 and 18 µL droplets could be transported at frequencies at 110 Hz and above on this configuration while 8 µL droplets would remain in place. Practically, this attribute could be extremely beneficial for a user-friendly device, as it would allow a large tolerance for the application of droplets by a user that does not have access to precision pipetting equipment. Furthermore, the higher reproducibility of step size on the dual-plate systems could meet applications requiring precise positioning of droplets.

The similarity in ARC threshold and velocity profiles between ARC-ARC and ARC-FOTS configurations indicate that pinning forces acting on the top and bottom plates of the ARC-ARC configuration do not sum. This effect is likely a result of the anti-phasic nature of this system, as these pinning forces will be out of sync. However, if only one plate were moving (e.g., their separation distance was increasing and decreasing), then these phases would be synchronized between the top and bottom plates. This configuration may provide for a lower ARC threshold and could potentially transport droplets vertically, but the changing plate separation would also complicate the formation of a seal to prevent evaporation.

Compared to an open ARC system, a dual-plate configuration expands the functional frequency range and normalizes device performance with regard to droplet volume. The dual-plate configuration also provides a simple means of sealing this system to reduce evaporation rates and offers new opportunities to improve the capabilities of the ARC system—particularly the ability to handle fluids with lower surface tension than pure water, control droplets with smaller volumes, and perform new functions (i.e., droplet fission) that have yet to be achieved on ARC systems, which will be the focus of future work. Overall, these results demonstrate the increasing potential of ARC systems to meet a wider variety of applications in microfluidics.

## Figures and Tables

**Figure 1 micromachines-10-00069-f001:**
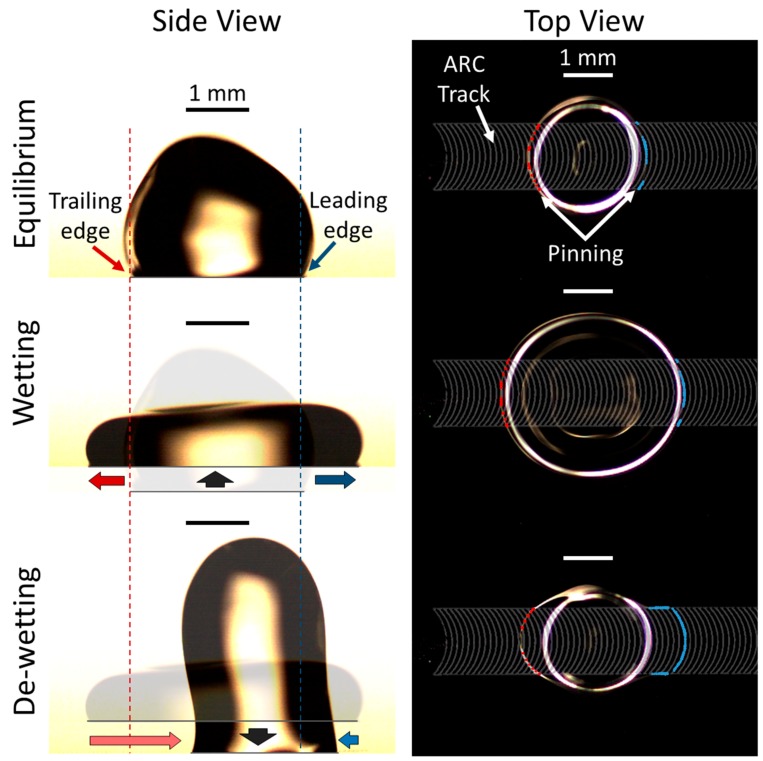
Mechanisms of droplet transport on anisotropic ratchet conveyors (ARCs). Droplets are transported by an ARC pattern (**top view**) that is composed of periodically spaced, curved rungs. The shape of the rungs creates a difference in pinning forces between the leading and trailing edges of the droplet, as only the front edge of the droplet can conform with the rung curvature. Vibration (vertical movement of the substrate) causes the edges of the droplet to oscillate between wetting and de-wetting phases. The increased pinning on the leading edge of the droplet creates an asymmetric force during the de-wetting phase, which results in a net force in the direction of droplet transport over a complete vibration cycle (**side view**). On the side view, the blue dashed line and arrows indicate the initial position and movement of the leading edge, and the red dashed line and arrows indicate the initial position and movement of the trailing edge, respectively. On the top view, blue and red dots indicate regions of the leading and trailing edges, respectively, that are pinned to the hydrophilic ARC pattern. Scale bars (black on side view and white on top view) indicate 1 mm.

**Figure 2 micromachines-10-00069-f002:**
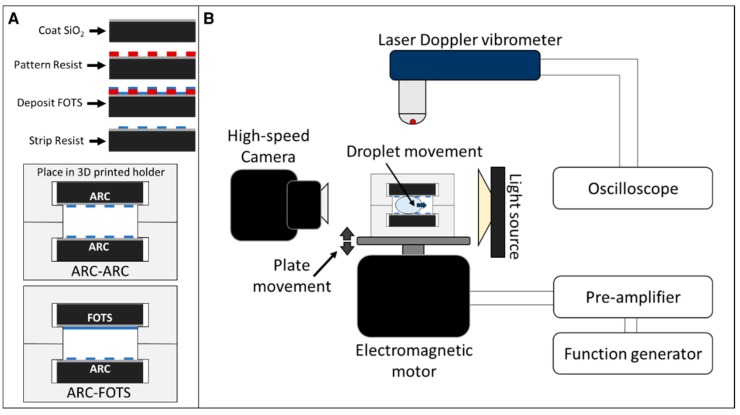
Fabrication process and experimental set-up. Anisotropic ratchet conveyors (ARCs) are fabricated by patterning resist on an oxidized silicon wafer, and coating exposed regions with a hydrophobic silane. (**A**) Stripping the resist reveals the hydrophilic ARC pattern. Wafers are diced into plates and secured in a 3D printed holder. (**B**) Droplet transport is examined by driving this system with an electromagnetic motor and function generator. The acceleration of the driving vibration is measured with a laser Doppler vibrometer and droplet motion is recorded with a high-speed camera.

**Figure 3 micromachines-10-00069-f003:**
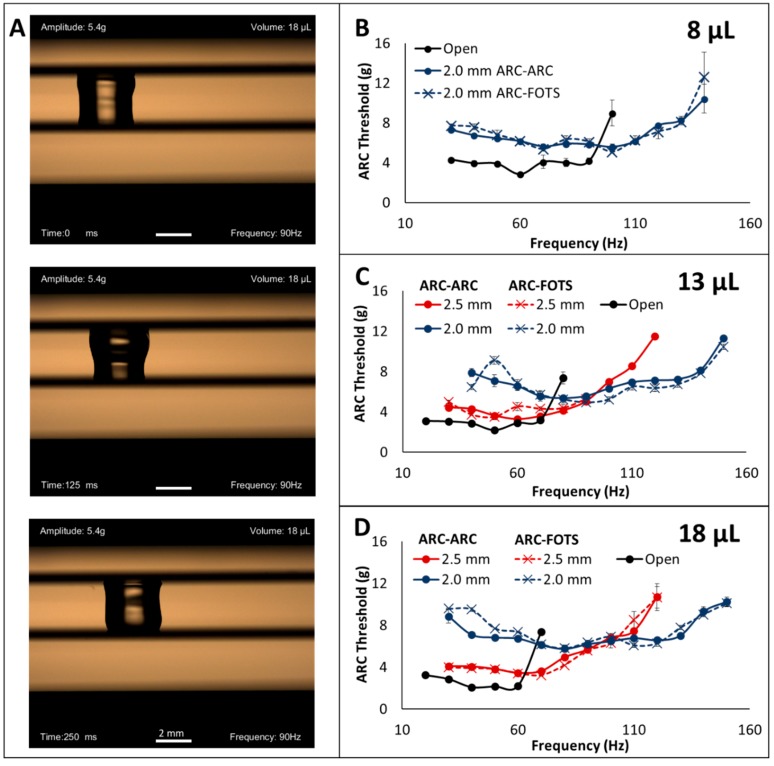
Anisotropic ratchet conveyor (ARC) threshold is dependent on separation distance with a dual-plate system. Droplets transported between two plates (**A**) exhibit a higher ARC threshold at low frequencies, but lower ARC threshold at higher frequencies, as well as a larger functional frequency range (**B**–**D**). This trend also repeats as the separation distance between plates is reduced. These observations are consistent for both ARC-ARC and ARC-FOTS configurations and all droplet volumes. Data are provided for measurements of ARC thresholds on three ARC devices and error bars represent ± standard deviation. Scale bars show 2 mm on droplet images.

**Figure 4 micromachines-10-00069-f004:**
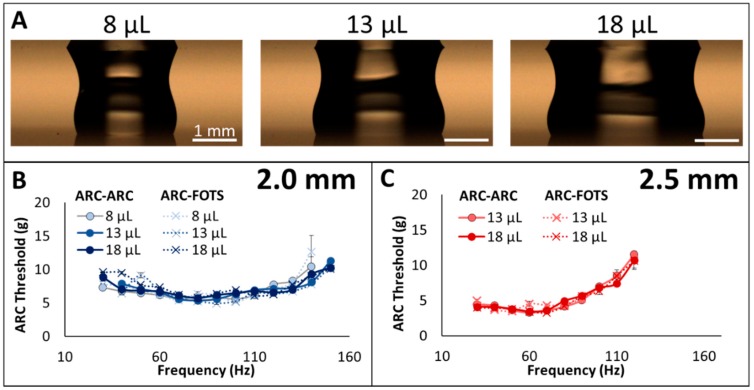
Anisotropic ratchet conveyor (ARC) threshold is independent of droplet volume in dual-plate system. ARC threshold values are similar for droplets of different volume (**A**) transported in 2.0 mm (**B**) and 2.5 mm (**C**) tracks. On open ARC systems, ARC threshold profiles are dependent on droplet volume [22], thus these profiles demonstrate that the ARC threshold profiles scale with vertical movement of the center of mass of the droplets. Error bars indicate ± standard deviation; scale bars on droplet images represent 1 mm.

**Figure 5 micromachines-10-00069-f005:**
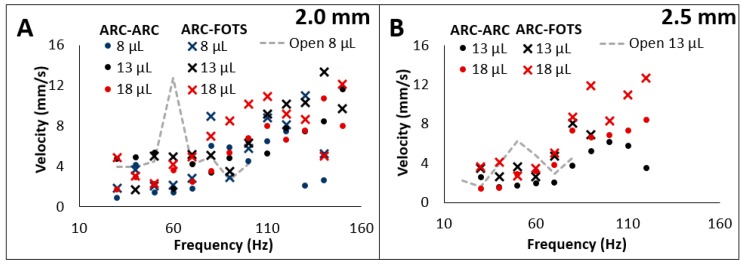
Transport velocity is correlated with vibration frequency on a two-plate system. Transport velocity at the anisotropic ratchet conveyor (ARC) threshold trends upward with increasing frequency on two-plate systems with (**A**) 2.0 mm and (**B**) 2.5 mm separations. Gray dashed lines indicate the velocity profiles of (A) 8 µL and (B) 13 µL droplets on open ARC systems with the same pattern. The velocity profiles on open systems are characterized by a peak velocity related to the droplet volume and a sharp drop in velocity at higher frequencies, whereas velocity profiles on the enclosed system generally increase with vibration frequency although volume dependent local maxima and minima are observed.

**Figure 6 micromachines-10-00069-f006:**
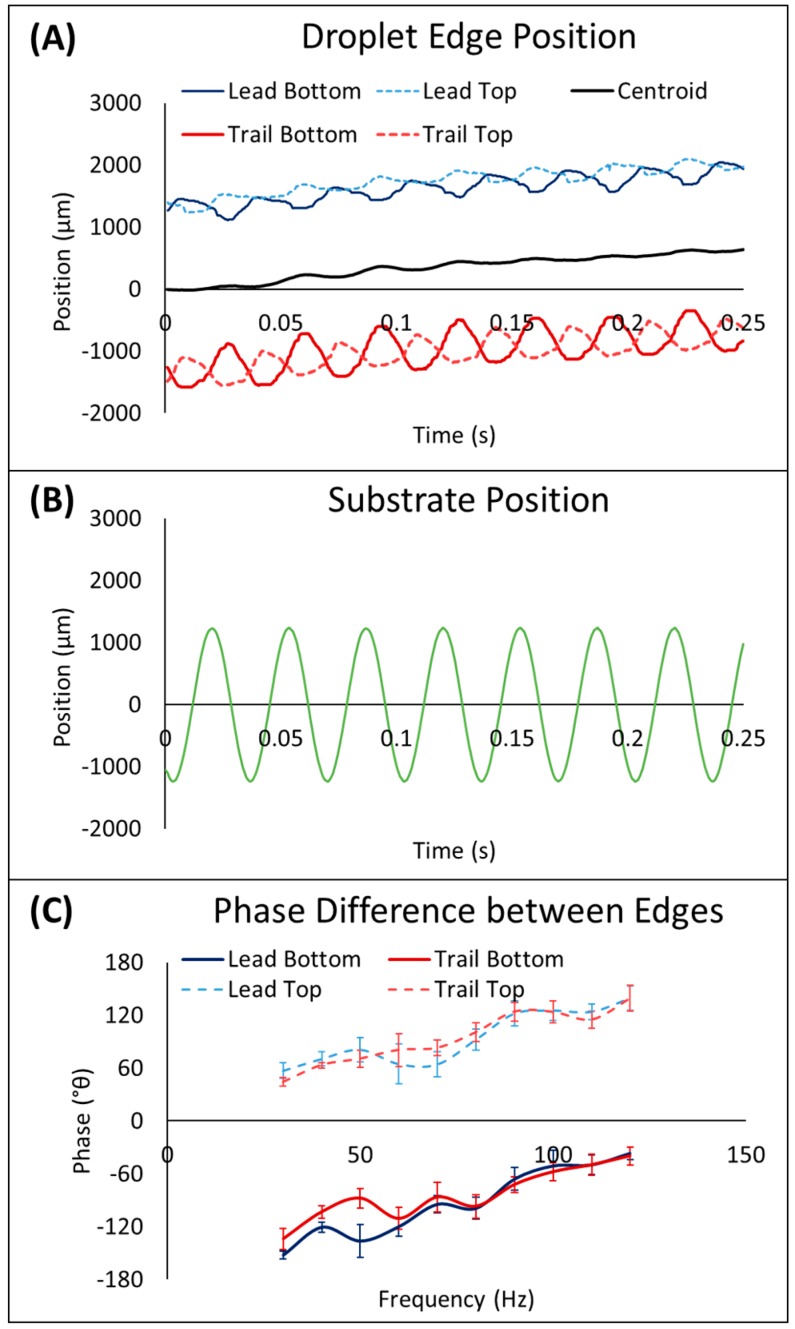
Droplet edges are anti-phasic between substrates. The wetting and de-wetting phases of contact line oscillation alternate between top and bottom edges (**A**). The movement of the droplet centroid also demonstrates that the ratcheting effect (i.e., small step backward and larger step forward) is still present for droplet transport on the enclosed system. Edge movement and the corresponding position of the driving substrate (**B**) is shown for a 13 µL droplet transported at 30 Hz in an enclosed ARC-ARC system with a separation of 2.5 mm. The phase difference, defined as the time delay between the highest position of the substrate and the furthest expansion of droplet edges, shows that top and bottom edges are anti-phasic (with a phase difference of 186.8° ± 16.4 between leading edges and 178.4°± 13.0 between trailing edges) throughout the entire functional frequency range for the system (**C**). Error bars show ± standard deviation of the phase angle for each frequency.

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
