# Peer review of "Vibration Induced Transport of Enclosed Droplets"

_micromachines, 2019, doi:10.3390/mi10010069_

Round 1
Reviewer 1 Report
The manuscript presents an extended work on the effect of a dual-plate enclosure system for the anisotropic droplet transport. Authors studied the influence of a top plate with anisotropic ratchet conveyor (ARC) patterns or a FOTS-coated hydrophobic flat top plate on droplet transport. And the dual-plate ARC shows the broader working frequency range than the open system. It can also normalize the transport performance of droplets of different volume in the device. Because of the anti-phase between top and bottom droplet edges, the droplet advances by the same numbers of rungs on ARC during each vibration cycle. Although many interesting experiment results shown in the paper, the analysis and explanation are that in-depth. I suggest the authors to address the following comments before the publication of this paper.
1. Please explain why the functional frequency range of dual-plate system (around 30-150 Hz) is broader than that of open system (around 20-100Hz).
2. The ARC threshold of open system and dual-plate system with 2.5 mm plate separation increases with the functional frequency, as shown in Fig. 3. But the ARC threshold of dual-plate system with 2.0 mm plate separation remains relatively stable throughout the functional frequency range, as shown in Fig. 4. Thus, why does the frequency have different effects on them?
3. I am curious why the droplet edge movement is anti-phasic on the enclosed system? Please elaborate on it.
4. The volume range of droplets which could be transported by the dual-plate system with the ‘volume-pass’ effect should be provided.
Author Response
Response to Reviewer 1 Comments
The manuscript presents an extended work on the effect of a dual-plate enclosure system for the anisotropic droplet transport. Authors studied the influence of a top plate with anisotropic ratchet conveyor (ARC) patterns or a FOTS-coated hydrophobic flat top plate on droplet transport. And the dual-plate ARC shows the broader working frequency range than the open system. It can also normalize the transport performance of droplets of different volume in the device. Because of the anti-phase between top and bottom droplet edges, the droplet advances by the same numbers of rungs on ARC during each vibration cycle. Although many interesting experiment results shown in the paper, the analysis and explanation are that in-depth. I suggest the authors to address the following comments before the publication of this paper.
Point 1: Please explain why the functional frequency range of dual-plate system (around 30-150 Hz) is broader than that of open system (around 20-100Hz).
Response 1: The addition of the top-plate restricts the motion of the droplet center of mass, which emulates the motion of smaller droplet (as smaller droplets can be driven at higher frequencies). Interactions with the top plate are also in play as well, so future work will look at capturing these complexities with a comprehensive model. This explanation was added to the discussion and conclusion section.
Point 2: The ARC threshold of open system and dual-plate system with 2.5 mm plate separation increases with the functional frequency, as shown in Fig. 3. But the ARC threshold of dual-plate system with 2.0 mm plate separation remains relatively stable throughout the functional frequency range, as shown in Fig. 4. Thus, why does the frequency have different effects on them?
Response 2: The smaller separation of the plates reduces the vertical motion of the center of mass of the droplet, emulating the motion of a smaller droplet (as noted above). However, contact of the droplet to the top-plate also affects the droplet behavior. The smaller separation of the plates would therefore provide the center of mass motion of a small droplet, but the surface contact of a large droplet. We hypothesize that this combination of effects is the cause of the stable ARC threshold with the 2.0 mm separation. While we aim to create a comprehensive model to elucidate all of these effects in the future, we believe that our hypothesis is currently too speculative and this complexity is outside of the scope of this paper, in which we have aimed to provide an empirical report of this system and these observed behaviors to stimulate new ideas and interesting questions such as this.
Point 3: The ARC threshold of open system and dual-plate system with 2.5 mm plate separation increases with the functional frequency, as shown in Fig. 3. But the ARC threshold of dual-plate system with 2.0 mm plate separation remains relatively stable throughout the functional frequency range, as shown in Fig. 4. Thus, why does the frequency have different effects on them?
Response 3: In the dual-plate configuration, the droplet edges are wetting on the top plate and de-wetting on the bottom plate as the droplet center of mass moves upward and vice-versa as the center of mass moves downward. This statement was added to section 3.3 to clarify the reason for this phenomenon.
Point 4: The volume range of droplets which could be transported by the dual-plate system with the ‘volume-pass’ effect should be provided.
Response 4: This volume range is dependent on the plate separation distance and the feature size of the ARC pattern, and therefore could be selected for specific applications. This clarification and a specific example relating to observations on the 2.5 mm plate system was added to the discussion and conclusion section of the manuscript.

Reviewer 2 Report
Article is well written and useful even without a good treatment of the mechanism for the change. The observation of consistent motion with each cycle could be useful for controlling the position or speed of the object without needing to have sensing and feedback control. It may be helpful to discuss these possibilities.
Figure 5: It would benefit from higher frequency in the open consideration and two-plate modes for more clarity. The limited data for the open configuration decreases the confidence in the conclusions of a local peak.
It would be appropriate to explicitly identify the fact that open platform is easiest to move across the range of conditions studied even though there is a region where the closed configuration is just lower acceleration compared to open under certain frequencies.
Author Response
Response to Reviewer 2 Comments
Point 1: Article is well written and useful even without a good treatment of the mechanism for the change. The observation of consistent motion with each cycle could be useful for controlling the position or speed of the object without needing to have sensing and feedback control. It may be helpful to discuss these possibilities.
Response 1: The is a very interesting possibility and the reproducible movement of the droplets on the dual-plate systems could meet applications requiring precise positioning. This possibility was added to the discussion and conclusion section.
Point 2: Figure 5: It would benefit from higher frequency in the open consideration and two-plate modes for more clarity. The limited data for the open configuration decreases the confidence in the conclusions of a local peak.
Response 2: This is a fair criticism, however droplets in the open configuration are not transported at higher frequencies. The provided plots for the open configurations cover the entire functional frequency range for these volumes. The limited frequency range of droplets in the open configuration is better demonstrated in Figure 3, wherein a sharp increase in ARC threshold is observed at the upper limit of the functional frequency range. The ARC threshold is so large at higher frequencies in the open configuration that the droplets will break apart before being transported.
Point 3: It would be appropriate to explicitly identify the fact that open platform is easiest to move across the range of conditions studied even though there is a region where the closed configuration is just lower acceleration compared to open under certain frequencies.
Response 3: This point was made more explicit in section 3.1.

Round 2
Reviewer 1 Report
The authors have made according revision and the paper is suitable for publication in its current format.